# Variation in personality can substitute for social feedback in coordinated animal movements

Isaac Planas-Sitjà [1✉], Jean-Louis Deneubourg[2] & Adam L. Cronin[1]

Collective movements are essential for the effective function of animal societies, but are complicated by the need for consensus among group members. Consensus is typically assumed to arise via feedback mechanisms, but this ignores inter-individual variation in behavioural tendency ('personality'), which is known to underpin the successful function of many complex societies. In this study, we use a theoretical approach to examine the relative importance of personality and feedback in the emergence of collective movement decisions in animal groups. Our results show that variation in personality dramatically influences collective decisions and can partially or completely replace feedback depending on the directionality of relationships among individuals. The influence of personality increases with the exaggeration of differences among individuals. While it is likely that both feedback and personality interact in nature, our findings highlight the potential importance of personality in driving collective processes.

[1] Department of Biology, Tokyo Metropolitan University, Tokyo, Japan. [2] Center of Nonlinear Phenomena and Complex Systems (CENOLI) - CP 231, Université libre de Bruxelles, Bruxelles, Belgium. ✉email: iplanass@pm.me

Decision making is a frequent and fitness-critical behaviour in most animals. In group-living organisms this is complicated by a potential conflict of interests among group members, which nonetheless must collectively reach some form of consensus if the group is to act effectively while remaining together[1]. A combination of empirical and modelling studies has demonstrated that these complex collective actions observed in many group-living organisms can be explained by individuals following simple local rules coupled with mechanisms of feedback[2–8]. The broad success of feedback-based models in explaining the dynamics of collective actions in a wide range of taxa has led some to suggest that mechanisms such as this may be common to a wide range of group-living organisms from social insects to primates[9,10].

In seeking to explain the complexity of animal collective behaviour, modelling approaches have often drawn from particle physics[4], and focused on the influence of interactions while assuming that group members are identical entities. However, inter-individual variation is a central tenet of natural selection[11], and it has long been recognised that variation in behavioural propensity among individuals underpins the successful function of complex societies such as those of social insects[12,13]. Furthermore, the recent abundance of studies in the field of 'behavioural syndromes' or 'animal personality' highlights the importance of this variability[14–18]. Behavioural variability among group members can be driven by differences in age, sex, condition or dominance status, among other things[19–21]. While not without controversy (e.g.,[22]), we use the term 'personality' here for reasons of word economy, to describe consistent individual variation in behaviour between individuals in a group, regardless of underlying cause of the variability. The available data indicate that the distribution of personality in a group can influence the outcome of various collective actions[18,23–31]. For example, colonies of honeybees containing behaviourally more diverse individuals are collectively more successful foragers than less diverse colonies[13] and groups with different mixed composition of shy/bold or social/asocial animals perform better than homogeneous groups in various taxa[31–33]. Personality differences can also lead to differences in how individuals contribute to decision making.

For example, personality differences can give rise to leader–follower patterns in pairs of sticklebacks with bold fish emerging as leaders and shy fish emerging as followers in experimental pairs[34]. Furthermore, these differences can be enhanced by social feedback: bold leaders may inspire faithful followership, and shy followers facilitate effective leadership[28,34]. It is therefore likely that personality differences within groups interact with feedback mechanisms to influence the outcome of collective decisions.

One context in which collective decision making is crucial is group movements. These actions are essential for groups to track resources or evade threats while maintaining group cohesion[35]. Collective movement decisions are thought to arise through mechanisms of positive feedback based on active signals[9,36–38]. For example, whooper swans (*Cygnus cygnus*) increase honking frequency until a threshold (quorum) number of individuals are calling before the flock takes flight[39]. Swarms of honeybees are stimulated into movement by the piping sounds of scouts who have already reached a consensus on a suitable site to relocate to[40]. Movement decisions may also be cue-based, for example, if the departure of one individual from a group itself indicates a desire to initiate a collective movement[41]. This latter mechanism follows a basic sequence of events: one individual initiates a movement and the rest of the group may or may not follow this initiative. The effectiveness of the initiative then depends on the propensity of other individuals to follow, and that of the initiator to persist in the action. Thus, the initial movement will become a collective movement only if a number of other individuals follow the initiator[42]. These movements are characterised by an all-or-nothing, U-shaped, pattern of the number of responders following the initiator[10,42–44]. Thus, initiations that are not joined are common but unsuccessful, those joined by many individuals are common and successful, while intermediate numbers of joiners are rare (see Fig. 1). This kind of decision process has been observed in fish[45,46], birds[47] and mammals, including primates[9,30,38,48], cattle[49,50] and Prezwalski horses[51].

Using a combination of modelling and natural observations, researchers have shown that these collective movements could be generated using two complementary feedback mechanisms,

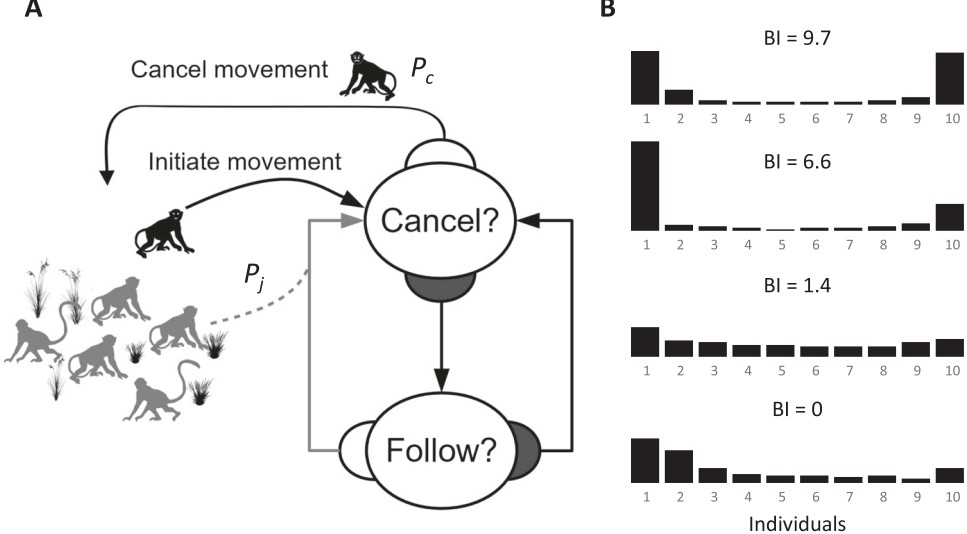

**Fig. 1 Flow diagram of the model and examples of bimodality index. A** Flow diagram of the model. In the decision nodes, dark-grey = no; white = yes. Initiator (black) starts a movement by departing from the group, and other monkeys (grey) can either join the movement or remain with the rest of the group. **B** Four examples of bimodality index (BI) for the probability distribution of the number of individuals following a given initiation. Higher index values indicate higher degrees of bimodality, and a BI of zero means that some of the prerequisites are not met (see methods). The U-shaped response arises because initiations typically either fail when they are not joined, or succeed when joined by all group members, while responses in between these extremes are rare.

echoing the process of collective decision making in social insects[10,38,52]. The first of these is a mimetic process in which the propensity to follow depends on the number of individuals already following. The second is that the propensity of the initiator to give up is negatively associated with the number of followers[10,50]. These mechanisms are consistent with patterns observed in many species with shared decision processes[10,38,53]. However, most of these previous studies have excluded the potential effects of variability among individuals. Recent models of self-organised processes indicate that personality variation among group members[27,43] and social affiliation between individuals[53–55] are important for explaining the emergence of collective patterns. For instance, several authors have shown that, against theoretical expectations, some individuals can be more successful initiators[42,50,53] or more inclined to follow[52,54] than others during collective decisions. The degree to which these differences in behavioural tendencies may influence group decision making and the potential interaction between feedback mechanisms and personality variation remains largely unexplored.

In this study, we use a theoretical approach to examine the potential importance of personality variation in collective decision dynamics by modelling movement decisions in small animal groups. Specifically, we explore the relative importance of inter-individual variation in personality and different feedback mechanisms on the effectiveness of collective movements, thorough exploring the following themes: (i) how the distribution of personalities within the group influences decision making, (ii) whether personality effects can operate in a top-down (despotic) and/or bottom-up (democratic) manner and (iii) the influence that social feedback between initiator and followers play in decision making. We test the hypothesis that within-group variation in personality alone is sufficient for collective decisions to emerge, and can reproduce the U-shape response pattern characteristic of collective decisions in various organisms. We also assess the veracity of our model by comparison of results with data from a field study of collective movements in Capuchin monkeys.

## Results

### Simulation model development

*Outline of model.* We use an agent-based model to investigate the decision-making process during collective movements. We initially focus on small animal societies, as these are common to a wide range of taxa and simulation results can be compared with available field data. Specifically, we model the process of collective movements of groups of ten agents by adapting a model that Petit et al.[10] used to simulate collective movements in Capuchin monkeys (*Cebus capucinus*). We first outline their model and subsequently explain how this was modified to incorporate personality effects. Petit et al.[10] considered that collective movement decisions arise through two mechanisms of feedback. The first one of these affects the probability of joining ($P_J$) a given initiation, while the second affects probability of cancelling ($P_C$) this initiation. Specifically, $P_J$ is a function of the total number of individuals in the group ($N$) and the number of individuals already departed ($R$):

$$P_J = \frac{1}{\alpha + \beta \frac{N-R}{R}} \qquad (1)$$

Petit et al.[10] used constants $\alpha = 162.3$ and $\beta = 75.4$, which were inferred from field observations, that reflect the mimetic behaviour among individuals. Thus, the probability of joining, $P_J$, increases with the number of individuals already departed, while the probability of the initiator cancelling, $P_C$, depends on the

following equation:

$$P_C = \frac{\delta}{1 + \left(\frac{R}{\gamma}\right)^e} \qquad (2)$$

where the $P_C$ of the initiator depends on a constant cancellation rate ($\delta$) and decreases with the increasing numbers of responders ($R$). The constant $e$ can be understood as an amplification effect depending on the influence of the individual who has joined the movement and $\gamma$ is the threshold. Petit et al.[10] obtained the best fit to observed data using values of $\delta = 0.009$, $\gamma = 2$, and $e = 2.3$. If $\gamma$ and $e = 1$, there is no amplification effect and thus any individual joining a movement will have the same effect on the initiator's $P_C$ (see below).

Following the same premises, the general flow of our model is as follows: first, at $t = 0$, a random individual (the initiator) advertises its desire to move by departing the immediate vicinity of the group. We consider only a single initiator per movement for simplicity, though field observations of monkeys suggest that simultaneous initiations by multiple individuals are rare[10,42,56]. This corresponds to situations where the initiation probability is small compared to $P_J$ and thus the time between initiations is greater than the duration of a collective movement. At subsequent time steps, the initiator has a probability of cancelling the movement $P_C$, while the remaining individuals (referred to as responders or followers) decide whether or not to join the movement with a probability $P_J$ (see Fig. 1).

Simulations were run for a period of 15 min with a timestep of one second. At each timestep, we updated individual behaviour based on the $P_C$ of the initiator and $P_J$ of followers. Simulations were halted before the 15-min time limit if all responders joined the movement or if the initiator cancelled the movement. Following each initiation, we scored (i) whether or not the movement was successful (some or all individuals joined the initiator) or failed (initiator cancelled the movement) and (ii) how many individuals had joined the initiator at the time of cancellation (or at the end-time). While we used groups of ten individuals, the same process can be extended to groups comprising any number of individuals (see analytical model section).

*Integration of personality.* To integrate personality variation into movement dynamics, we remove the feedback function in Eq. 1 so that it becomes dependent on a single parameter $\alpha'$:

$$P_J(r) = \frac{1}{\alpha'}; r = 1...N \qquad (3a)$$

where the $P_J$ of a given responder $r$ within a group of $N$ individuals depends on a single parameter $\alpha'$. This can be understood as a parameter defining the magnitude of a personality trait[27,43,53], which influences collective movement decisions by altering the individual probability of joining, with higher values associated with lower joining probability. In our model, the parameter $\alpha'$ has no effect on the probability of initiating a movement. All individuals thus have the same probability of becoming an initiator, which aligns with the available empirical data[10,38,39,49–51].

*Despotic and democratic scenarios.* To understand how the distribution of traits might influence collective processes, it is necessary to elucidate how collective phenomena emerge from individual actions under different leadership scenarios[57]. We consider here that collective decisions can arise in a top-down (despotic) or bottom-up (democratic) manner. In despotic decisions, one or a few individuals assume the role of the leader and decide(s) for the rest of the group[36], while democratic decisions

typically emerge in a self-organised manner[6,7,58,59]. In the context of our model, the parameter $\alpha'$ could determine both an individual's social influence (its effect as an initiator on the propensity for others to follow) and the individual's response threshold (its propensity to follow others).

From a top-down (despotic) perspective, the parameter $\alpha'$ of the initiator determines the $P_J$ of other group members and Eq. 3a becomes:

$$P_J(r) = \frac{1}{\alpha'_i} \qquad (3b)$$

where the $P_J$ of a responder $r$ depends on the $\alpha'$ of initiator $i$. Thus, when highly influential individuals act as initiators, they will have a strong influence on others (other individuals have a high and identical $P_J$), generating a rapid response from other group members, and resulting in a high probability of initiating a successful movement (making them effective leaders). On the other hand, low influence initiators will have more difficulty in attracting other individuals, and a low probability of initiating a successful movement (making them poor leaders).

Under the democratic scenario, the joining propensity $P_J$ for each individual depends on its own $\alpha'$, and is not influenced by the identity of the initiator. Individuals with low $\alpha'$ (high $P_J$) will readily join movements regardless of the identity of the initiator, while individuals with high $\alpha'$ have a low probability of joining movements. Equation 3a, thus becomes:

$$P_J(r) = \frac{1}{\alpha'_r} \qquad (3c)$$

where the $P_J$ of a responder $r$ depends on its own $\alpha'$, no matter who initiates the movement. This situation corresponds to some individuals being more eager to follow movements than others, as observed in[52].

**Modelling different distributions of personality.** Animal personality can be considered a binary character state or a continuum depending on the trait in question, but presently there is little understanding of how such characters might be distributed within groups[60,61]. The type of distribution (e.g., uniform, normal or bimodal) may have strong implications for the emergence of collective behaviours such as decision making[62–65].

To model the influence of the distribution of personalities in the group, we thus divided our simulations into two conditions. In the first, which we refer to as the unimodal condition (Uni), values of $\alpha'$ were drawn from a truncated normal distribution. In the second, which we refer to as the bimodal condition (Bim), values of $\alpha'$ are drawn from two separate truncated normal distributions, one of which has a higher mean than the other. Thus, in the unimodal condition, individual personalities are randomly distributed over a continuum of values, representing variability within a single class of individuals. This is in line with studies showing some individuals recruit followers faster or have more social bonds than others[50]. In the bimodal condition, groups are comprised of individuals with high or low $\alpha'$, which is more representative of societies with distinct bold/shy or dominant/subordinate classes of individual[66].

To test how the degree of inter-individual variability affected movement decisions, we proceeded as follows: in the case of Uni, we gradually expanded the range of the distribution of $\alpha'$ from an initial value of 160 (in accordance with[10]) by values of 100 in both directions with maximum and minimum values of 10 and 4460. In the Bim scenario, the standard deviation of both normal distributions was fixed at 100 and the mean for low $\alpha'$ individuals remained at 160, while the mean value of the high $\alpha'$ distribution was increased by values of 100, to a maximum 3960. In all cases, we start with a set of simulations in which all individuals have the

same value of $\alpha'$, representing a control state without personality for each scenario. In our Bim scenarios we also modelled the influence of different leadership structures through modifying the proportion of high:low $\alpha'$ individuals, by varying the number of individuals selected from each of the distributions (1:9 … 9:1).

*Initiator's propensity to cancel.* Previous studies have indicated that the cancellation rate of the initiator ($P_C$) is crucial for effective emulation of natural group patterns[10,38,53]. We consider four mechanisms that could underlie the decision of an initiator to cancel a movement. The first, as used in[10], is Eq. 2 described above, where each responder influences the $P_C$ of the initiator in a non-linear way. We refer to this as the non-linear $P_C$ condition. The second mechanism (linear $P_C$ condition; Eq. 4) supposes that the addition of followers ($R$) influences the cancellation rate ($C$) of the leader in a linear manner instead of a quorum-like response like in Eq. 2, and this influence is proportional to the group size ($N$):

$$P_C = C\left(1 - \frac{R}{N}\right) \qquad (4)$$

When considering a linear $P_C$, we replace the cancellation rate $\delta$ with $C$ (=0.004). As we do not consider any kind of feedback, $C$ needs to be lower than $\delta$, and this value was obtained by computing the mean value of Eq. 1 for $N = 1...5$. Note that the results obtained from Eq. 4 are qualitatively the same as those obtained from Eq. 2 with $\gamma = 1$ and $e = 1$.

Under the third mechanism (constant $P_C$; Eq. 5):

$$P_C = C \qquad (5)$$

individuals have a fixed cancellation rate $C$[63], identical for all individuals and determined by their internal motivation state, which is independent of the number of followers or size of the group. Finally, we consider that the $P_C$ of initiator $i$ is related to the personality of individual $i$, that is, proportional to its $\alpha'$:

$$P_C(i) = C(\alpha'_i); i = 1...N \qquad (6)$$

Thus, we hypothesise that individuals with lower $\alpha'$ will have a lower rate of cancelling. We refer to this as the $C\alpha'$ condition. In these last two cases (Eqs. 5 and 6), we exclude any positive feedback between responders and initiator.

*Simulations overview.* The above scenarios are modelled in a full-factorial manner, the principal components of which are summarised in Fig. 2. Thus, we consider two cases where the $P_C$ of the initiator is under the influence of positive social feedback (non-linear and linear cancellation type), and two cases where there is no social effect (constant and $C\alpha'$). The control simulations (without personality) for each scenario were performed with the corresponding $P_C$ while maintaining the same $\alpha'$ value ($P_J$) for all individuals. The control scenario for the non-linear condition, for instance, therefore corresponds to simulations with Eq. 3 ($\alpha' = 160$) and Eq. 2, while for constant and $C\alpha'$ conditions we used the most basic assumption: Eq. 2 ($\alpha' = 160$) and Eq. 5. These four cancellation types were modelled for each expanding range of $\alpha'$, distributed either unimodally (Uni) or bimodally (Bim), and from a despotic (top-down) and democratic (bottom-up) perspective.

*Assessing model performance.* We use two approaches to assess the efficacy of each scenario and parameter combination in replicating movement patterns typical of natural groups. First, we assess the bimodality of the distribution of the frequency of different numbers of responding individuals for each combination of parameter values (see Methods for details). Bimodality of the response distribution can be considered indicative of an all-or-nothing type response typical of collective decisions in various

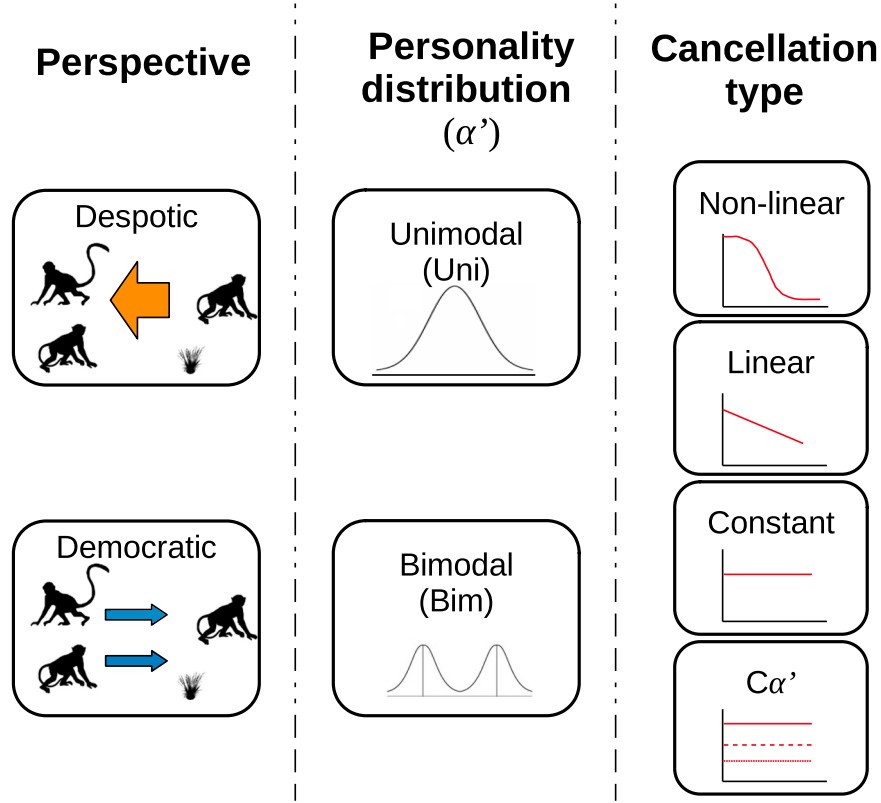

**Fig. 2 Summary of the components of our full-factorial simulation analysis.** Perspective determines whether the propensity for potential followers to join a movement depends on the personality of the initiator (despotic) or their own personality (democratic). Personality distribution defines the distribution of personality types in the group. Cancellation type models how an initiator's propensity to give up depends on the number of followers.

organisms[10,44,67]. With respect to our model, bimodal response distributions are expected for simulation sets in which movements involving only a single individual or all ten individuals are common, but responses involving intermediate numbers of individuals are rare (Fig. 1B). That is, bimodality indicates the preponderance of two classes of outcome: failed initiations (when no movement occurs) and successful movements by the whole group.

Second, we compared simulation results to published field data of collective movements by Capuchin monkeys[10,42]. Capuchin monkeys are arboreal primates that form small groups (4–40 individuals) of multiple males/females of different ages and must collectively move between foraging sites frequently (see Methods for more details). The dataset used by Petit et al.[10] comprised 255 collective movements of a group of Capuchin monkeys, which has been the focus of a series of theoretical and observational studies[10,42,43], including the original model in Eqs. 1 and 2. In order to compare simulation results to field data, we quantify the differences between the distribution of responders (e.g., Fig. 1B) obtained from simulations and from field data by means of the Manhattan distance.

### Simulation results

*Assessment of bimodal response patterns.* We observed pronounced variation response patterns between Bim and Uni conditions and democratic and despotic scenarios. Overall, the despotic scenario shows bimodal patterns of the number of responders more often than the democratic scenario. Under Uni conditions, bimodal response patterns were observed for both despotic and democratic scenarios only when considering low or no personality variation ($\alpha'$ values ranging from ~10 to 1000),

coupled with non-linear (or occasionally linear) feedback (Fig. 3A, B).

Under Bim conditions (Fig. 4A), response patterns in despotic groups were highly dependent on the number of leaders (individuals drawn from the low $\alpha'$ distribution). In groups with few leaders, only weak bimodality was observed, and this was mostly at low levels of personality variation and with feedback present. The highest degrees of bimodality were observed for despotic–Bim scenarios with a balanced proportion of leaders and followers. Of these scenarios, those with the greatest differences in the two distributions of $\alpha'$ and incorporating social feedback produced the most bimodal response patterns. Interestingly, however, scenarios with personality but without feedback mechanisms ($C\alpha'$) also resulted in bimodal response patterns over a large range of parameter values (Fig. 4A), indicating that such patterns can arise from within-group differences in personality alone. With high numbers of leaders, weak to moderate bimodal response patterns could be produced with or without social feedback. Higher bimodality was observed with increasing distance between the distributions of $\alpha'$.

For the democratic scenario, bimodality was rare, and observed only in cases with low numbers of leaders, limited variation in $\alpha'$, and feedback in place. In such case, Uni condition (Fig. 3B) leads to bimodality for a wider range of parameters compared to Bim condition (Fig. 4B). No matter how personality is distributed in a democratic (bottom-up) system, increasing the personality variation leads to low rate of bimodality.

*Comparison to field data.* We compared the distribution of the number of individuals involved in each movement generated by our simulation to that observed for a group of Capuchin monkeys in the field. For Uni conditions, there was only weak matching

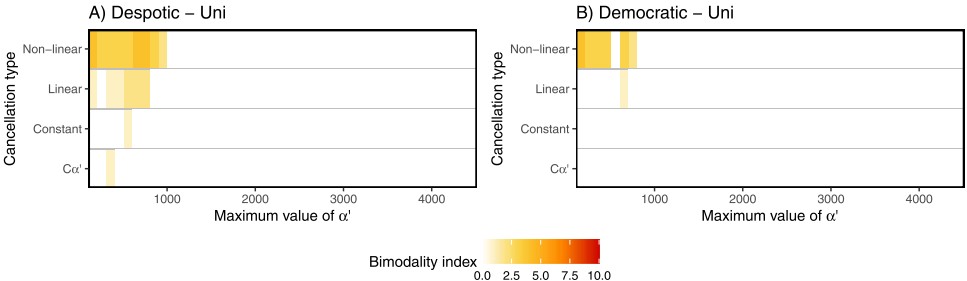

**Fig. 3 Scenarios where a bimodal distribution of the number of responders was produced under the Uni condition.** Scenarios where a bimodality was produced under **A** despotic or **B** democratic scenario, depending on the cancellation type (*y*-axis) and the range of difference in $\alpha'$ values (*x*-axis). The first column of the *x*-axis corresponds to the control simulation without personality as $\alpha' = 160$ for all individuals. For the following columns the minimum value of $\alpha' = 10$ and the maximum value of the distribution is given by *x*-axis. The probability of joining per second (one timestep) is equal to $1/\alpha'$. The colour scale indicates the bimodality index (see Methods for computation details); the higher the index, the higher the degree of bimodality (see also Supplementary Figs. S3 and S6).

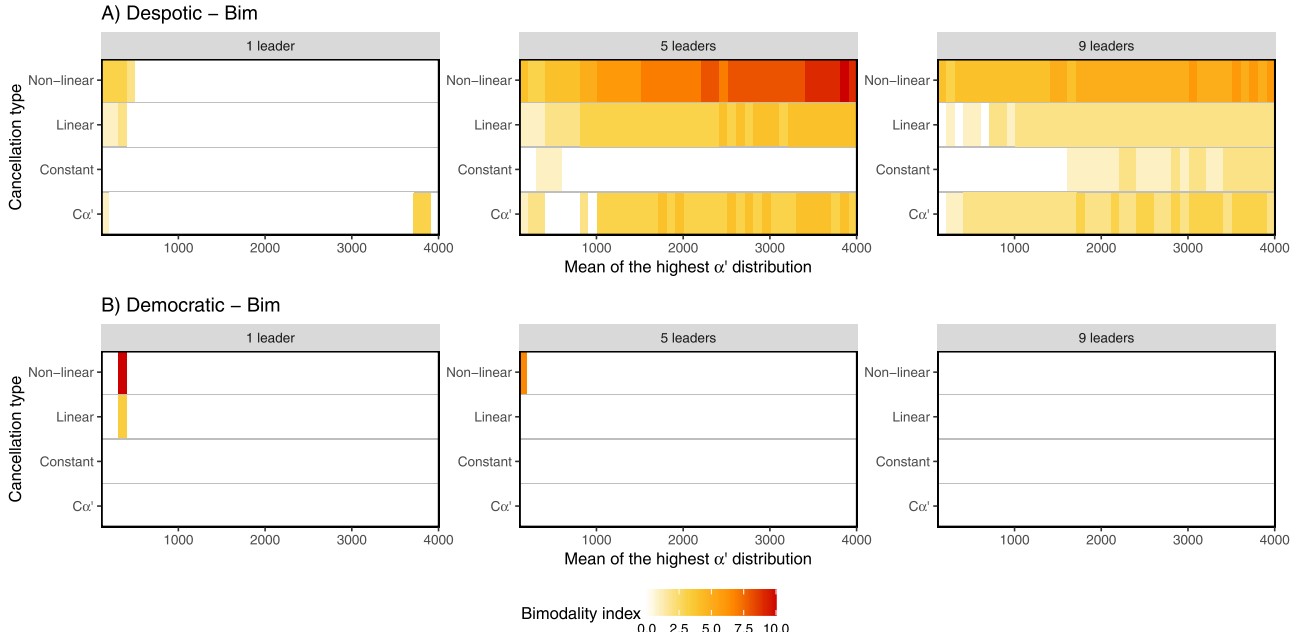

**Fig. 4 Scenarios where a bimodal distribution of the number of responders was produced under the Bim condition.** Scenarios where bimodality was produced under **A** despotic and **B** democratic scenarios. *Y*-axis indicates the cancellation type and the *x*-axis is the range of difference in $\alpha'$ values. For each condition, we show results for simulations with 1, 5 or 9 leaders (see Supplementary Figs. S2 and S5 for more results). The first column of each graph corresponds to the control simulation without personality as $\alpha' = 160$ for all individuals. For the following columns the mean of the low $\alpha'$ distribution was fixed at 160, while the mean of the highest $\alpha'$ distribution is shown on the *x*-axis. The probability of joining per second (one timestep) is equal to $1/\alpha'$. The colour scale indicates the bimodality index (see Methods for computation details); the higher the index, the higher the degree of bimodality (see also Supplementary Figs. S1 and S4 and Supplementary Movies 1 and 2).

between simulations and field data, with the closest comparisons being found for simulations with non-linear feedback and low-mid variability in $\alpha'$. Notably, however, the best-fitting parameter combinations for simulations with personality but no feedback ($C\alpha'$) achieved a similar degree of fit to the field data as the results of feedback-based simulations in both despotic and democratic scenarios (Fig. 5).

Simulations under Bim conditions generally produced a better fit to field data than Uni conditions, though this varied depending on the number of leaders, degree of variability in $\alpha'$ and between despotic and democratic scenarios. The broadest fit of simulations to natural patterns was produced under a despotic scenario with non-linear feedback (Fig. 6A). However, the despotic scenario with no feedback ($C\alpha'$) also closely modelled field data for

intermediate numbers of leaders. The democratic scenario generally produced a poorer fit to field data (Fig. 6B), though interestingly showed the greatest consistency with the despotic scenario for personality-only ($C\alpha'$) conditions.

As a further comparison between simulations and field data, we plotted frequency distributions of simulation outcomes against the field data of Petit et al.[10] for key simulation results (Fig. 7). The scenario Bim–despotic with a balanced proportion of leaders: followers (5:5) reproduced the distribution of responders involved in natural movement dynamics as well as a model considering identical individuals with both feedback mechanisms (Petit model) (Fig. 7A, B). Interestingly, a despotic scenario with no feedback ($C\alpha'$) closely modelled field data for particular numbers of leaders (see Figs. 5A and 7C). In fact, the scenario $C\alpha'$ did a

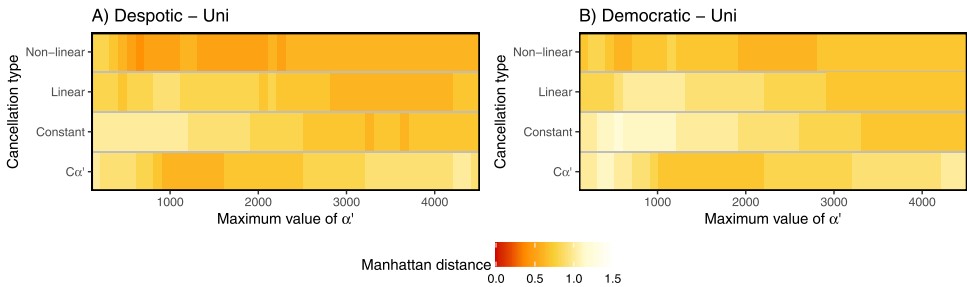

**Fig. 5 Manhattan distance between the distribution of the number of individuals involved in each initiation generated by our simulations and the one observed in the field for a group of Capuchin monkeys.** The Manhattan distance was obtained by computing the difference between field and simulation data for each bar of the histogram showing the number of individuals moving in each initiation. The lower the index of Manhattan distance, the greater the agreement between both distributions. **A** Results under a despotic scenario and Uni condition and **B** under a democratic scenario and Uni condition, both considering non-linear, linear, constant, and $C\alpha'$ types of cancellation rate (y-axis). The first column of the x-axis corresponds to the control simulation without personality as $\alpha' = 160$ for all individuals. For the following columns the minimum value of $\alpha' = 10$ and the maximum value of the distribution is given by the x-axis. The probability of joining per second (one timestep) is equal to $1/\alpha'$.

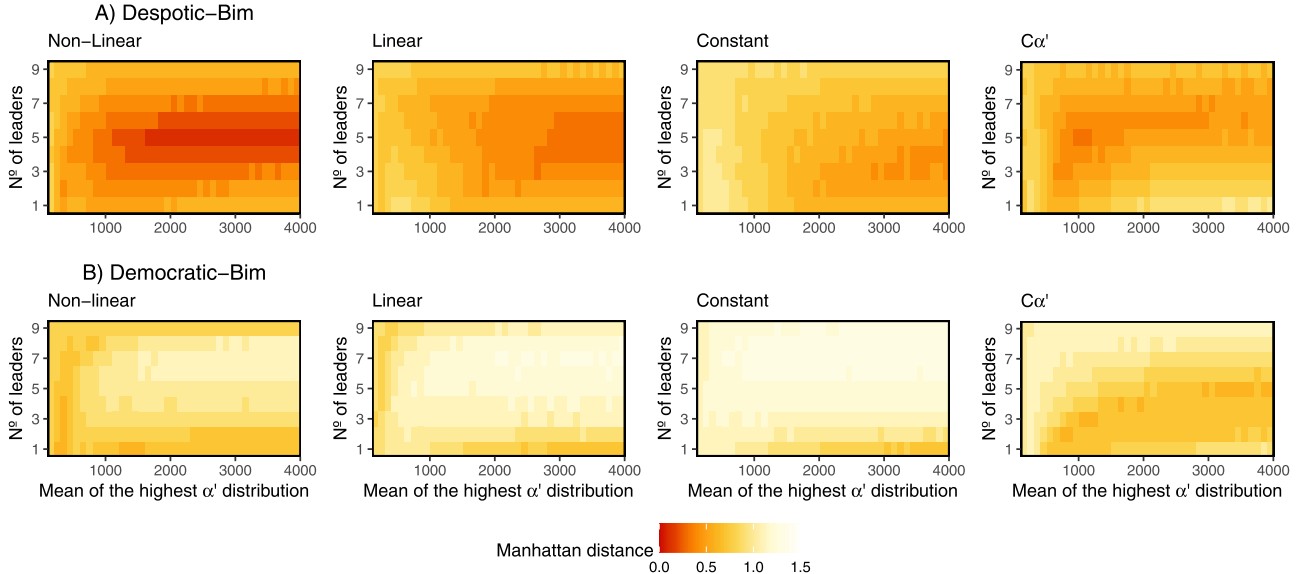

**Fig. 6 Manhattan distance between the distribution of the number of individuals involved in each initiation generated by our simulations and the one observed in the field for a group of Capuchin monkeys.** The Manhattan distance was obtained by computing the difference between field and simulation data for each bar of the histogram. The lower the index of Manhattan distance, the greater the agreement between both distributions. **A** Results under a despotic scenario and Bim condition and **B** under a democratic scenario and Bim condition, both considering non-linear, linear, constant and $C\alpha'$ types of cancellation rate. The y-axis indicates the number of leaders (number of individuals drawn from the lowest $\alpha'$ distribution) within a group. The first column of the x-axis corresponds to the control simulation without personality as $\alpha' = 160$ for all individuals. For the following columns the mean of the low $\alpha'$ distribution was fixed at 160, while the mean of the highest $\alpha'$ distribution is given by x-axis. The probability of joining per second (one timestep) is equal to $1/\alpha'$.

better job replicating the field data than the control models with individuals having the same rate of joining ($\alpha' = 160$) and with non-linear, linear or constant $P_C$ (Fig. 7D–F). This indicates that variability in $\alpha'$ alone can produce patterns equivalent to those typically assumed to be driven by feedback mechanisms.

**Analytical model**. As a final step, we sought to clarify the relative importance of feedback and inter-individual variability through development of an analytical model[68]. We used this to test the hypothesis that systems of identical individuals could never produce a bimodal (J or U shape) response distribution without feedback, or if limited to linear feedback. This model initially assumes that there is no feedback between initiators and followers, and thus that the cancellation rate is constant. The master equation, which represents the counterpart of stochastic multi-agent simulations above, is as follows (see Supplementary

Material for more details):

$$\frac{d\Theta(0,1,t)}{dt} = -(\Psi(0,1;1,0) + \Psi(0,1;1,1))\Theta(0,1,t) \quad (7a)$$

$$\frac{d\Theta(i,1,t)}{dt} = \Psi(i-1,1;i,1)\Theta(i-1,1,t) - (\Psi(i,1;i,0) \\ + \Psi(i,1;i+1,1))\Theta(i,1,t); i = 1,\dots,N-1 \quad (7b)$$

$$\frac{d\Theta(N,1,t)}{dt} = \Psi(N-1,1;N,1)\Theta(N-1,1,t) \\ - (\Psi(N,1;N,0))\Theta(N,1,t) \quad (7c)$$

$$\frac{d\Theta(i,0,t)}{dt} = \Psi(i,1;i,0)\Theta(i,1,t); i = 0,\dots,N \quad (7d)$$

The state of the system is described in terms of a probability function $\Theta(i, k, t)$. At time $t$, $i$ is the number of individuals who

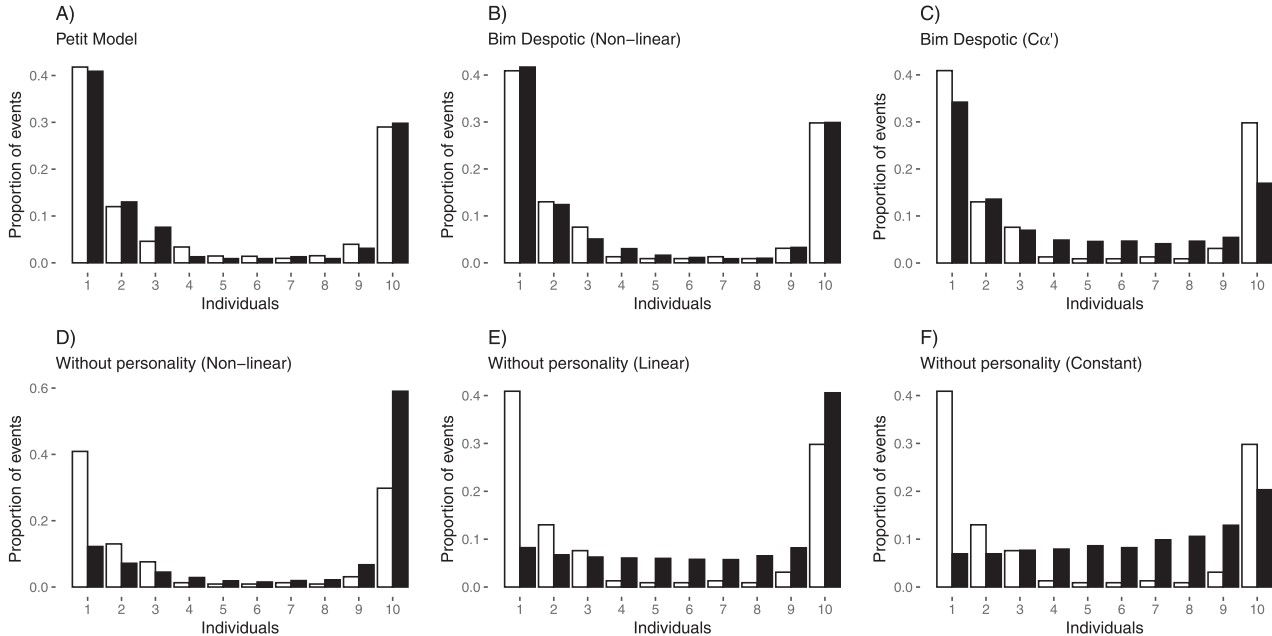

**Fig. 7 Distributions of individuals involved in each initiation for key simulations.** Comparison of distributions of individuals involved in each initiation for key simulations (black bars) versus field data of Capuchin monkeys (white bars[10]). **A** corresponds to the Petit model, including feedback on $P_C$ and $P_J$; **B** Bim–despotic scenario, with non-linear feedback, ratio of 5:5 leaders:followers and mean of the highest $\alpha'$ distribution of 3160; **C** Bim–despotic scenario, with C$\alpha'$ cancellation rate, ratio of 6:4 leaders:followers and mean of the highest $\alpha'$ distribution of 960. Scenarios without personality are depicted with **D** non-linear, **E** linear and **F** constant feedback.

have started to follow the initiator ($0 \le i \le N$, where $N$ is the total number of potential followers) and $k = 1$ if the initiator is still moving or $k = 0$ if the initiator cancelled its movement. Equation 7 describes the change over time of the probability that the system ($\Theta(i, k, t)$) occupies each one of a discrete set of states. We define state as the number of responders to a given initiation, with the initiator still on the move or having cancelled the movement. $\Psi(i, 1; i + 1, 1)$ is the probability of a transition from $i$ to $i + 1$ responders (a new individual joins the movement), while $\Psi(i, 1; i, 0)$ is the probability of transition from a state with $i$ responders to the state in which the initiator cancels the movement. These transitions depend on both the probability of joining ($P_J$), defined by the identity of the initiator (despotic system), and the initiator's probability of cancelling the movement ($P_C$).

*Constant probability of cancelling ($P_C$) and following ($P_J$).* In the case where the $P_C$ and $P_J$ of the initiator are constant:

$$\Psi(i, 1; i, 0) = P_C \tag{8a}$$

and with identical individuals, the probability of transition from $i$ to $i + 1$ responders is:

$$\Psi(i, 1; i + 1, 1) = P_J(N - i) \tag{8b}$$

with $N - i$ being the number of potential followers still at rest. Under these conditions, and with a time limit as specified for our simulations, the final distribution of group responses with Eq. 7 could never be bimodal. When extending the model to infinite-time, thus $\Theta(i, 1, \infty) = 0$, the total distribution of the number of followers $\Theta^T(i, \infty) = \Theta(i, 0, \infty)$:

$$\Theta^T(i, \infty) = \frac{N!}{(N - i)!} \frac{a^i}{\prod_{l=0}^{i}(a(N - l) + 1)} ; i = 0, \dots, N \tag{9a}$$

with $a = P_J / P_C$. In such case, the distribution of responders to a given initiation was never bimodal, although this distribution was either skewed towards higher (if $P_J > P_C$) or lower (if $P_J < P_C$) numbers of responders depending on the $P_J/P_C$ ratio. If $P_J = P_C$,

the distribution was flat (Eq. 9a). With infinite-time, $1/P_C$ and $1/P_J$ are respectively the mean duration of the movement and the mean duration before a follower joins the movement. Thus, the mean fraction of followers corresponds to:

$$<f> = \frac{a}{a + 1} \tag{9b}$$

In the case of personality, and with infinite-time, each initiator is characterised by its own value $a_k$ ($P_C$ and/or the $P_J$ differ among individuals). The global distribution of the number of followers is:

$$\Theta^T(i, \infty) = \sum_{k=1}^{m} f_k \frac{N!}{(N - i)!} \frac{a_k^i}{\prod_{l=0}^{i}(a_k(N - l) + 1)} ; i = 0, \dots, N \tag{10}$$

where $f_k$ is the fraction of initiations characterised by a value $a_k$.

Under a despotic system (Eq. 10), numerical analyses show that the minimum condition required to produce a bimodal outcome is to have a proportion of the group with a ratio $P_J/P_C < 1$ and another proportion with a ratio $P_J/P_C > 1$. This condition can produce a U-shape distribution of the number of responders to each initiation (maximum values at 0 and $N$, with the minimum value in between; see methods and Supplementary Figs. S7 and S8). Under a democratic scenario, there was no numerical solution giving a bimodal outcome (see Supplementary Material).

*Probability of cancelling ($P_C$) decreases with the number of followers (linear feedback).* Subsequently, we tested the behaviour of the analytical model when introducing a linear feedback in the initiator's $P_C$ with unlimited time:

$$P_{Ci} = \frac{P_{C0}}{1 + i} \tag{11a}$$

where $P_{C0}$ is the intrinsic probability of cancelling, and $i$ is the number of followers. In this case, and without differences between individuals, the probability of transition between state ($i$, 1) and

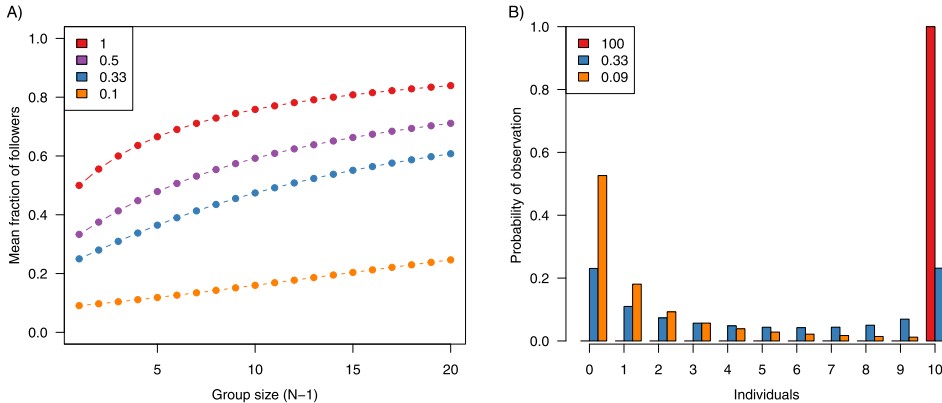

**Fig. 8 Results from the analytical model. A** Relationship between the mean fraction of followers (number of followers/$(N-1)$) and group size ($N-1$) for different values of $P_J/P_{C0}$. **B** Probability of obtaining $i$ followers (individuals) considering a group with ten potential followers ($N-1 = 10$). In both figures, colours indicate the ratio $P_J/P_{C0}$.

$(i, 0)$, which means cancelling a movement, is:

$$\Psi(i, 1; i, 0) = P_{Ci} \qquad (11b)$$

The probability of a transition from state $(i, 1)$ to state $(i + 1, 1)$ (i.e., an individual joins the movement and the initiator is still on the go) is:

$$\Psi(i, 1; i + 1, 1) = P_J(N - i) \qquad (11c)$$

For long $t$, thus $\Theta(ð, 1) = 0$ and $\Theta^T(ð, 0) = \Theta(ð, 0, \infty)$, the probability distribution of responders to a given initiation is given by:

$$\Theta^T(i, 0) = \sum_{k=0}^{i} C_{ik} \frac{P_{Ci}}{P_J(N - k) + P_{Ck}} \; i = 0, \ldots, N \qquad (12a)$$

with

$$C_{00} = 1 \qquad (12b)$$

$$C_{ik} = \frac{P_J(N - (i - 1))C_{i-1k}}{P_J(k - i) + P_{Ci} - P_{Ck}}; k = 0, \ldots, i-1; C_{ii} = -\sum_{k=0}^{i-1} C_{ik} \qquad (12c)$$

This model predicts that the mean fraction of followers increases with the total number of potential followers and the ratio $P_J/P_C$ (Fig. 8A), and as such is approximated by:

$$<f> = \frac{N^\gamma a^{\gamma'}}{1.3 + N^\gamma a^{\gamma'}} \qquad (12d)$$

with $\gamma \approx 0.6$, $\gamma' \approx 1.1$.

In addition, we analysed the probability distribution of followers ($\Theta^T(i)$) according to $a$ (ratio $P_J/P_C$). When the ratio $P_J/P_C$ was very large, the distribution of $\Theta^T(i)$ was monomodal, with individuals always following the initiation ($\Theta^T_N = 1$). When the ratio $P_J/P_C$ was very small, the distribution of $\Theta^T(i)$ was also monomodal: $\Theta^T_0 = 1$. These results are comparable with the results obtained without feedback in $P_C$ (Eq. 9a). Interestingly, between these two extreme situations, the distribution of $\Theta^T(i)$ can be bimodal, depending on the ratio $P_J/P_C$. If $P_J$ is larger than $P_C$, then $\Theta^T_N > \Theta^T_0$ and vice versa (Fig. 8B). These results indicate that at least some sort of feedback between initiator and responders is needed to generate a bimodal distribution when groups are composed of identical individuals.

Finally, we observe that the value of the ratio $P_J/P_C$ at which we observe the maximum bimodality decreases with the size of the group (see Supplementary Fig. S9). This means that when increasing the number of potential followers, the maximum

bimodality is achieved with higher differences between $P_J$ and $P_C$ values. The critical $P_J/P_C$ value decreases with about $N^{-0.6}$ (see Supplementary Fig. S10).

*Discriminating between personality and social effects.* The analytical solutions above provide the basis for the following methods to discriminate between response patterns driven by feedback (social interactions) or personality variation. To facilitate such discrimination, we propose the following three methods. By limiting ourselves to the final distribution of responses as above, and if we can identify individuals, the computation of the number of responders according to the identity of the initiator using a non-parametric test would show whether some individuals are more often followed than others. If the identity of individuals is unknown, but it is possible to carry out observations on groups of different sizes, the mean fraction of responders ($<f>$) is independent (or quasi-independent) from the total population size when individuals show personality variation and there is no feedback:

$$<f> = \sum_{i=1}^{l} q_i \frac{a_i}{a_i + 1} \qquad (13)$$

where $q_i$ is the fraction of initiations by individuals with the value $a_i$, with $a = P_J/P_C$.

If the information required for the above is not available, we can still discern between mechanisms through the study of dynamics. Let us consider two extreme cases; on the one hand two types of initiators with different $P_C$, and on the other hand a species without personality but with negative feedback influencing their cancellation probability $P_C$. The collective movement dynamics of the group can be divided among several states of $i$ moving individuals and $t_i$ (seconds) duration. The states with $i$ moving individuals can only end with the addition of a new responder or by the cancellation of the movement by the initiator. If the distribution of durations for $i$ responders corresponds to an exponential law, the average duration $<T>$ of the state of $i$ moving individuals for the species with feedback is:

$$<T> = \frac{1}{P_C(i) + P_J(N - i)} \qquad (14)$$

and the standard deviation is also equal to $<T>$. Therefore, the coefficient of variation (CV) is $= 1$. In the case of a species with

personality, the mean time <T> corresponds to:

$$<T> = \frac{1}{l_m} \sum_{l=1}^{l_m} T_l; T_l = \left( \frac{1}{P_{Cl} + P_{Ji}(N-i)} \right) \qquad (15)$$

$l_m$ being the number of initiators ($\leq N$), each with its own $P_C$. In such case, the variance is:

$$V = \frac{1}{l_m} \sum_{l=1}^{l_m} \left( <T>^2 - 2T_l<T> + 2T_l^2 \right) = -<T>^2 + \frac{2}{l_m} \sum_{l=1}^{l_m} T_l^2 \qquad (16)$$

$$CV = \frac{V^{0.5}}{<T>} \qquad (17)$$

If $T_l$ differs (i.e., $P_C$ or $P_J$ differ among individuals), the coefficient of variation will be larger than 1. In addition, a survival curve analysis of the state, which depends on the number of individuals already departed, can be used as a proxy to estimate whether we observe a feedback or personality-related process. If the survival function is different from an exponential law, the process is likely to be under the influence of personality. If, on the other hand, the distribution fits an exponential function, it is suggestive of a social feedback-based mechanism. Thus, the coefficient of variance, combined with the analysis of survival curves, could give us information on whether the process underlying a collective action is mainly driven by social feedback or personality differences. The supposition that CV = 1 in groups of identical (or almost identical) individuals is theoretical and is based on the assumption of a high number of observations without memory or learning effects. Thus, the analysis of natural observations might show some variation from the expectation of CV = 1. It may be possible to statistically test the deviance of experimental observations from a theoretical CV = 1 as follows. First, we compute the average duration <T> and CV. We then perform $n$ simulations, with $n$ being the number of observations, and compute the CV of those $n$ simulations for states of 0, 1, 2 … $N-1$ followers ($N$ being the group size). After repeating this process many times (e.g., 1000 or 10,000) we can generate a theoretical distribution of CV under the assumption that individuals are identical. If the experimental CV is not statistically different from this distribution, we can rule out personality effects.

## Discussion
Many theoretical studies have shown that signal and cue-based feedback mechanisms in a wide variety of group-living organisms underpin a range of collective actions, from the movement decisions of monkeys[69] to the foraging and emigration behaviour of large colonies of social insects[4,59,70]. However, few of these studies have incorporated inter-individual differences in behavioural propensity (i.e., animal personality), despite the growing evidence that this phenomenon is important in determining the outcome of collective actions[18,32,71,72]. Our results clearly show that while linear or non-linear feedback mechanisms can drive movement decisions in small groups of animals, the same collective patterns can arise without feedback mechanisms under certain scenarios if individuals vary in a single personality parameter. While previous studies have explored the effects of adding personality components to feedback-based models[27,43], our results are the first to our knowledge to demonstrate that personality can entirely replace feedback mechanisms in collective actions. While it is likely that in nature both of these mechanisms interact to generate the patterns of collective behaviour, this highlights the potential importance of personality as an alternative explanatory mechanism, where feedback-based responses are typically assumed to hold the predominant role. This is

particularly pertinent considering the wealth of recent evidence for the existence of personality in diverse taxa, from those living in small groups[73–76] to large self-organised societies[21,26,71].

While our results show that both despotic and democratic scenarios could generate a bimodal distribution of moving individuals, the despotic scenario was effective at doing so over a broader parameter space. Furthermore, only the despotic scenario could model natural groups effectively in the absence of feedback (i.e., $P_C$ constant and $C\alpha'$). The democratic scenario produced a bimodal response distribution only when considering non-linear feedback and low inter-individual variability (democratic–Bim/Uni with non-linear $P_C$). These results, together with the mathematical model, suggest that personality alone cannot give rise to effective collective actions when driven from the bottom-up. On the other hand, the despotic scenario was effective over various combinations of parameter values without feedback in place. This may indicate that democratic systems are likely to rely more on feedback in collective decision making, whereas despotic systems can function without. It is notable that increasing differences between individuals reduces the relative importance of feedback in both scenarios (Uni/Bim) and that these feedback become necessary to replicate natural patterns when considering identical individuals. Indeed, a bimodal response pattern never arose in the absence of feedback with identical individuals in simulations, and this was confirmed by our analytical model (see also[10]). Based on these theoretical findings, we speculate that large societies such as social insects, often considered exemplary democratic systems, should rely strongly on feedback and exhibit limited influence of personality. On the other hand, we might expect to find a larger influence of personality in small societies, even without well-established social hierarchies, where shared decisions may be driven by top-down, despotic, mechanisms. Quantifying individual variability in societies of different sizes and with clear democratic or despotic systems during collective decision making would help to test this prediction.

Our simulations indicate that bimodal outcomes were produced over a broader range of parameters by groups with a bimodal distribution of personalities than under Uni conditions. The same was true of the match between simulation outcomes and the natural patterns of collective movement in Capuchin monkeys. It is worth noting that there is a fundamental difference between Bim and Uni conditions, in that sets of individuals drawn from the Bim distribution will always have bimodally distributed personalities, while those drawn from the Uni distribution will form different distributions of $\alpha'$ each time. We might thus expect consistently bimodal patterns in the collective movements of Bim groups, while Uni groups will give rise to different response distributions in each simulation depending on the distribution of $\alpha'$ values. The success of our Uni scenario in producing bimodal patterns under some conditions may thus be in part an artefact of replication, in that different parts of the distribution of responses will be emphasised in each replicate, which in combination can generate the bimodal response pattern. This may thus be a poor approximation of success on a case-by-case basis, as might be expected in nature. Thus, the limited success of unimodally distributed personalities we find may actually overstate the effectiveness of this distribution of personalities, adding further weight to our finding that bimodality enhances success.

The fact that bimodal distributions of personality most effectively approximate the behaviour of natural groups with or without feedback is perhaps not surprising. Several studies have shown that groups composed by heterogeneous personalities, or a mixture of phenotypes, perform better than homogeneous groups[13,31–33]. Furthermore, several theoretical models predict a stable bimodal distribution of leadership[23,65,77] depending on

density, context and benefits associated with each strategy[65,78]. In addition, our results indicate that the best approximations of natural patterns were found in groups in which there was not only a bimodal distribution of personalities, but also an approximate numerical balance in personality types (a ratio of 5:5 leaders:followers, or 7:3 in $C\alpha'$). This is in agreement with a recent model suggesting that balanced distribution of personalities can increase the rate of successful movements in small groups[43,79]. The fact that groups with balanced, bimodal distributions of personality can better replicate natural dynamics raises an important question: does this mean that groups adaptively tend towards such compositions to optimise collective actions? If so, we might expect regulation mechanisms to exist which facilitate compositions of distinct leaders and followers or different extreme phenotypes. Optimal distributions could be a product of selection against individuals within groups with ineffective compositions, which either fail or disband as a result[19,24,80]. Alternatively, adjustment of personalities to the optimal composition could arise through social conformity and behavioural plasticity[81] or through admittance or exclusion of new individuals by other group members based on their suitability for forming an optimal distribution of personalities[24].

Our findings indicate that personality is a potentially important component of collective behaviour, and can act synergistically with, or possibly replace, mechanisms of positive/negative feedback. In our model, these effects were an emergent property of variation in a single personality characteristic. This characteristic could represent various behavioural traits, such as boldness[27], motivational state[29,30], social status[53] or even vocalisation intensity[82,83] and as such, our findings may be applicable to a broad range of group-living organisms. Future studies could explore the influence of additional personality parameters, as this may lead to synergistic or stabilising effects resulting in more complex group-level responses. It would also be valuable to assess the relative importance of personality and feedback mechanisms in generating collective behaviours in different taxa with differing social systems. To this end, we demonstrate several methods that could help elucidate the relative importance of personality and social effects during collective actions, even when the identity of individuals is unknown. We suggest that discrimination between these effects requires data with accurate timing of events, and can be facilitated by using marked individuals where possible. This framework will help improve our understanding on the relative importance of personality variation, positive/negative feedback, social bonds and hierarchies in generating collective behaviours in different kinds of group-living organisms and help answer the question of which, if any, of these components is a universal driver of collective behaviour.

## Methods

**Subject of study**. Capuchin monkeys are arboreal primates who form persistent groups of 4–40 individuals[84] composed of multiple males and females of different ages, and with high degree of relatedness within groups. Females usually remain in the same group, while males tend to migrate to new social groups[85]. These monkeys forage on several types of plants and trees, looking for fruits and small animals or insects[85], and must collectively move between foraging sites frequently. The experimental data used in this study come from a semi-free-range group of monkeys established in 1989 at the Primatology Centre of the Louis Pasteur University, Strasbourg, France. This group was subject of several experimental and theoretical studies[10,42,43]. The group was composed by five males aged of 2, 6, 6, 7 and >20 years old and five females; two of 5, one of 7 and two of >20 years old. Individuals of more than 20 years old were wild-born, while others were born in captivity.

**Simulations**. Simulations were performed in Python 2.7[86]. We considered 16 scenarios in total (Fig. 2). For Bim/Uni we considered different ranges of variability: 117 different ranges of $\alpha'$ for Bim (160: 3960) and 132 for Uni (10: 4460). For Bim (despotic/democratic) we considered different proportions of follower:leaders (high:low $\alpha'$) within groups[9] for the full range of $\alpha'$. We performed

5000 simulations for each unique combination of parameters using seconds as unit of time. In total we performed 11,850,000 simulations (2370 unique parameter combinations).

**Bimodality index**. In order to classify a distribution of a collective movement of $N$ individuals as bimodal, the frequency values for the extremes of the distribution (1 and $N$) should be higher than those for intermediate values, as these extremes respectively correspond to situations where the initiator cancelled after attracting no followers, or was successful in initiating a group movement (all individuals moved). There are several indexes to statistically determine if response distributions could be considered bimodal[87]. To stress the differences between the frequency of the extreme values (1 and 10 followers), we calculated a bimodality index (BI) as follows. First, we checked that the extremes (in this case 1 and 10 responders) correspond to the two maximum values. Second, we took the lower number of observations of the two extreme values of the distribution (1 and 10) and calculated the ratio between this and the mean number of observations for all other number of responders (i.e., 2 to $N-1$ agents). Therefore, an index of 2 means that the lowest of the two extreme values of the distribution (either 1 or 10) was twice the mean of the frequencies of all other numbers of responders [2–9 agents] (see Fig. 1B). For the analytical model section, the computation of BI was simplified and we used:

$$BI = \left( \frac{A_s - A_m}{A_s} \right) \tag{18}$$

where $A_s$ is the amplitude of the smaller peak of the two extreme values of the distribution (i.e., 0 and 10), and $A_m$ is the amplitude of the minimum value of the distribution[88].

## Data availability

The data that support the findings of this study are available within the paper and Supplementary Information. Additional data are available on request from the corresponding author.

## Code availability

The simulation code can be found in the Supplementary Information.

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

## Acknowledgements

This study was funded by Japan Society for the Promotion of Science (JSPS) and Tokyo Metropolitan University.

## Author contributions

I. P.-S. and J.-L. D. conceived the original idea. I. P.-S. wrote the first draft of the manuscript, carried out the simulations, analyses and graphical representation. A. L. C. helped develop the original idea, supervised the project and contributed to the final manuscript. J.-L. D. developed the analytical model and contributed to the final manuscript.

## Competing interests

The authors declare no competing interests.
