## [Peer Review File · Communications Biology]

Reviewers' comments:

Reviewer #1 (Remarks to the Author):

This paper addresses a fascinating issue – the role of individual differences in personality (here, in tendency to follow or to be followed in group movements) in producing collective leader-follower dynamics that resemble a set of observed movements by capuchin monkeys where initiators tended either to have no followers or to get all individuals in a group of 10 to follow. The authors' analysis of an agent-based model shows that while feedbacks can generate the observed bimodal distribution of following, under some conditions, individual differences in tendency to be joined could also generate this pattern. This is a very interesting insight, though the specific condition when it occurs might actually not be very interesting (see my final comment below). My main problem with the paper, however, is that it leaves too much poorly explained. Thus a reader who is interested in following the model, as opposed to simply accepting the main result could have great difficulty. To me, the paper should only be accepted for publication after a major revision to make the model and the presentation of the results more comprehensible.

First, a small note: personality variation usually refers to consistent individual differences in behavior - consistent over time or across contexts. This model only requires that for each event, individuals differ. It is not clear that it requires that those differences persist across time or contexts.

Figure 1b shows the distribution of proportion of 'events' involving different numbers of individuals, where the text and figure legend provide just enough information for readers to know that this represents outcomes for capuchin monkeys. I suggest describing the scenario in a bit more detail in the text; e.g., how many movements were observed? What the definition of an initiation and a join? What was the sex/age composition of the group (i.e. how much variation in leader-follower tendencies were likely given the natural history of the system)?

Equations (1) and (2) need more explanation. Lines 135-141 sort of explain these two equations, but do not do a good job. Equation 1 seems to say that the probability of a given individual j joining an initiator depends on two parameters: α which affects the general tendency to join, and β which affects the strength of response to the number of other followers. Equation 1 has no personality variation. In contrast, equation (2) potentially has individual differences in a_i , but no feedback. Because equation (2) says that P_j depends on a_i and not a_j , it appears that there is individual variation in how strongly individuals attract others to join, but no individual differences in tendency to join? Or, because the equations are not well explained, I was not confident that this is what the authors have in mind. Later, when the authors describe Despotism and Democratic Scenarios, they appear to be saying that a_i could either represent variation in tendency to be joined (Despotic) or in tendency to join (Democratic). If this is what the authors have in mind, they need to explain this more clearly and explicitly when they first introduce equations (1) and (2). The text also mentions an α' which does not appear in either equation. The text implies that the authors may have a 3rd equation that has both the feedback and individual differences, but they do not show that equation, or is equation (2) meant to be somehow included in equation (1) in the simulation? Perhaps that is what the authors mean to say, but this is not clear.

The authors then assumed that the distribution of tendencies to attract others to join (or do they mean tendency to join) was either uniform or bimodal. I suppose this is OK, but certainly can be criticized since my sense is that more often personalities are normally distributed. Can the authors cite any references with data that distributions of tendencies to join or to attract others to join is uniform or bimodal?

They then have equations where the initiator's probability of canceling is lower if it has more followers (a feedback mechanism), or where P_c is proportional to a_i . Some small points to clarify: In equation (3), γ is undefined – what is its biological meaning, and in equation (6), the authors should note whether a_i is Despotic or Democratic.

Figure 2 seems to show that in their simulations, personality variation is uniform or bimodal, implying no analyses where there is NO personality variation, but I thought the authors would be

comparing outcomes with vs without personality variation, where the latter would instead just have feedbacks. I also thought they would have simulations with personality variation but no feedbacks on joining, but this does not appear to be represented in figure 2.

Figure 3 shows the Manhattan distance between simulation and experimental histogram data. While this is no doubt a useful way of quantifying fit of a model to data, the figure itself is incomprehensible. The x-axis is either Distance or Maximum, and the y-axis runs from 1-9 with axis labels that are largely unclear as to their meaning. I could guess as to what they might mean, but it would just be a guess. I can see from the color patterns that the model tended to fit the data better in some boxes than others, but how this relates exactly to the 3 brief paragraphs of text under Democratic scenario is unclear. I could probably figure it out, but it would take significant work. I am not against working to understand a paper, but it would certainly be better if the authors made this less difficult for readers. I suggest labeling every panel of figure 3 with a letter and then referring readers to compare particular panels to see the points that the authors wish to make. As a small note, the heading 'Democratic scenario' seems inappropriate since the results also include the Despot scenario.

Finally, one could view it as unsurprising, almost trivial, that if individuals differ substantially in their tendency to be joined – in particular if this tendency is bimodally distributed, then the result could be a bimodal distribution of outcomes where some individuals that tend to not be followed are not followed, while those that have a strong tendency to be followed are indeed followed. The authors note that this result is unsurprising on line 362. Fortunately, in the lines following line 362, the authors suggest some interesting ideas that go beyond their simulation's somewhat obvious result.

Although it could be viewed as beyond the scope of this paper, it would certainly be interesting to see not only whether the different situations generate bimodal joining results as with one set of observations on capuchins, but also to report on what WERE the outcomes of other simulated scenarios? How do feedbacks and individual differences affect other outcomes of interest?

Reviewer #2 (Remarks to the Author):

Please see attached file

*** copied in by editor ***

Planas-Sitjà, Deneubourg & Cronin present a factorial modelling analysis investigating the role of personality in generating patterns of collective behaviour, and make comparisons between the patterns created through model simulations with those seen in a data set of capuchin monkey movement initiations.

At the heart of this paper is an important idea: while observed patterns of collective decision making (in particular, 'U-shaped distributions') have been attributed to social feedback rules, might these patterns actually arise more from variation between individuals? Investigating this hypothesis is valuable as (in my opinion at least), researchers are often too quick to attribute collective patterns to a putative social feedback rule because the two are consistent, without considering alternative explanations.

However, despite this interesting core idea, I found the paper difficult to parse and ultimately unconvincing.

The predictions of all models are compared quantitatively to one data set from a capuchin monkey study system, and yet little to no pertinent information about the study system is given when

considering the plausibility of these models. For example, despotic decision systems are introduced by way of ref 36, which contains no mention of capuchin monkeys. Is there any evidence that capuchins might have a despotic system? I appreciate this might be unknown (in which case testing different models would be useful for understanding that study system), but comparison of the large factorial design against data from one study system seems ill-suited to answer wide-ranging questions about the role of personality in generating collective behaviour. As the authors note, U-shaped distributions as in Fig 1 are common across many study systems, and this is a qualitative rather than quantitative feature that models of collective decision making should reproduce. Therefore I would have found it more useful simply to know which models (and parameters) could produce a U-shaped distribution, rather than the quantitative fit to a single study system (I could not judge precisely whether the data were newly collected from a previously established system or whether the data were the same as used in a previous study). As far as I can tell, the data is acting as a proxy for the generally desirable U-shape, since the goal is to make general inferences about collective decision making, rather than to explain the specific mechanisms behind capuchin groups.

It took me a long time to work out exactly what features of the data were being compared to simulations. These are described opaquely in the manuscript, e.g. SI line 24: 'To compare patterns generated by simulation and empirical data': what 'patterns'? It would be much clearer to say exactly 'To compare the distributions of the number of agents involved in each event' or similar. Adding a figure showing some exemplar histograms of this distribution generated from different models would then make Fig 3 much easier to understand. Although these histograms are an important feature of the data and of collective behaviour, I don't find a match between a model and the data on this one feature to be a convincing case that collective behaviours attributed to social feedback could actually be due to personality variation. There are many other ways that such systems could be explored, such as more detailed time sequences of following/abandoning, how often specific individuals are followed or lead, time intervals between decisions etc. For example, it is quite clear that a despotic model with bimodal leaders should produce a U-shape, since initiators are either very strongly followed or barely followed (by definition). This could be straightforwardly tested: do initiations succeed only under specific individuals? This is key since these despotic-bimodal systems are predominantly those that are good fits to the data. I would have liked to see either: (i) a case made that social feedback could not be distinguished from personality variation across a wide range of possible observations or (ii) a clear delineation of what observations *can* distinguish these, as a recommendation for future empirical practice.

In conclusion, I don't think this paper represents a sufficiently substantial contribution for publication in Communications Biology as it focuses on a quantitative fit to one study system, and on a relatively limited view of how patterns of collective behaviour can be described.

Minor comments:

Eqn 2, the α_i should presumably have an associated prime/apostrophe

Fig 3. The meaning of the x-axes is not given in the caption here as it is in eq. Fig S1. The colour scale is somewhat unhelpfully labelled simply as 'values'

Fig 3 and elsewhere: the scale of the alpha parameter is presumably based on the 1s intervals of the simulations, but this is very unclear, especially as one must reach L410 before seeing that this is how the simulations were done. I think alpha values should be given units (seconds?) for clarity.

The ms notes that the results obtained using the Manhattan distance are qualitatively the same as the KS or the BC. However, it is noteworthy that differences between parameters are much clearer in Fig S2 than in Fig 3

L230 as with the point above, the scale of these alpha variation values is unclear. Furthermore, what does "variability in α " mean? Is it standard deviation, distance between modes, range?

Eqn 6 and elsewhere: the notation P_{ca} is very unclear: I think the alpha here is simply to remind

us that this P_c depends on α prime, but it looks like a multiplicative factor.

L410 Simulations: Simulating at fixed 1s intervals is probably fine, but in principle using the Gillespie algorithm would be a better approach, and much more efficient.

Reviewer #3 (Remarks to the Author):

In this current manuscript the authors investigate through computer modelling if differences in personality would properly describe previously published experimental data of white-faced capuchins initiating (with or without success) a collective moment of the whole group.

This dataset acquired by Petit and collaborators was previously reproduced by using an statistical model which altered the probability of a conspecific joining an initiator depending on the ratio of already decided monkeys, while taking into account that the initiator itself could abort its attempt to trigger collective movement in the group.

The authors here have implemented changes to that previously developed model, which currently takes into account differences in personalities (i.e. chances of joining an initiator or aborting an attempt). The authors have succeeded in proposing an intriguing combination of personal characteristics, which they related to different group level organizations, such as "democratic" or "autocratic" relations between the agents.

My criticism of this article is focused mainly on its form, but also on the lack of some simple (and in my opinion needed) additional analysis and comparisons of their models with trivial cases in the main text.

After checking the submission guidelines of COMMSBIO, I noticed that perhaps the suggested (or enforced?) structure of the article caused the authors to come up with a structure that is very confusing for the reader. In normal circumstances, even if the fine details of the model were left for a later methods section, a more in depth description of the model should be given at the "outline of the model section". As is, that section is just confusing and the reader has to scroll through the end of the article to read all the methods section before proceeding further.

Still related to this, Eq 1 and 2 are just randomly dropped in the text, with mentions on their meaning, significance and use coming later. Unless there is a specific guideline for equations to be treated as "floating figures without a label", equations should come after a text explaining them, or at least introducing what follows. Additionally, the presentation of the previous results and modelling was lacking, and only contained a shallow description of their results. For example, the model description or results makes no mention on how the original P_j changes over time as more agents join the initiator. Such comments are in the SI and I think they should be added to the main text.

The shallow explanation given by the authors up to this point continues to cause problems for the reader later on: the definitions of "probability of joining" $P_j = \alpha_i$ are misleading, since from equation 2 it is assumed that an agent J joins the group solely based on the characteristic of the initiator I. While this is true for the despotic scenarios, on the democratic scenarios the equation is $P_j = \alpha_i$. Changes to the subscript of the equation should be made to account to all possible scenarios in the study, e.g. " $P_j = \alpha_k$, where $k=i$ or j depending on despotic or democratic scenarios" and so forth.

After jumping ahead to the methods sections I was left with the current understanding of the model:

- 1 - An agent is sorted to become an initiator (uniform distribution? never properly described in the text)
- 2 - This attempt on triggering a move is active either until the initiator aborts it (T_a), or the time of the simulation runs out ($T_f=900$ time steps which is equivalent to 15min).
- 3 - Agents test their own probabilities P_j of joining the initiator based on their specific scenario

3b - It is not clear if the attempt succeeds in case all other agents join an attempt before Tf.

A similar (but more in depth) analysis is given in the theoretical part of the SI, tackling how the trivial cases of the model would work, but given it's brevity and SI placement the main article suffers from not having it.

Figure 3 I'm sorry to say it's abysmal. Color resolution (specially to color blind readers) makes it very hard to follow. Inverting the color scheme (such as Figure S2) would solve problems for even the most extreme cases of color blindness. The caption is lacking in information, where again SI figures have the minimal information for a reader to follow them. Furthermore, with the lack of caption, the x and y axis of each subplot are basically a mystery until the reader reads the appropriate label for figures S1 and S2.

Overall the work is sound, the research valid, but the total experience seems like a proper manuscript was reduced to its bare minimum to fit a size limitation which I am not aware it exists. My suggestion is for drastic changes to be made in the form of the article, properly explaining the model and simulations before the results, and more importantly, properly describing the limiting cases and theoretical analysis that are only found on the SI. My review was leading to complaints about a similar analysis (concerning average time before an initiator aborts it's behavior), when I noticed similar and more in depth analysis were made in the SI but just briefly introduced in the main text, where they are sorely needed.

We have thoroughly revised the manuscript. This includes the following major changes:

- A complete restructuring of the presentation to move details from the methods and supplementary material to the main text.
- A refocussing of the presentation of our results to first demonstrate matching of simulation outputs to theoretical predictions (bimodality of response patterns) before additionally comparing results to field data
- An expansion of the analytical model to cover additional possibilities, and integration of this into the main text.

Reviewer #1 (Remarks to the Author):

This paper addresses a fascinating issue – the role of individual differences in personality (here, in tendency to follow or to be followed in group movements) in producing collective leader-follower dynamics that resemble a set of observed movements by capuchin monkeys where initiators tended either to have no followers or to get all individuals in a group of 10 to follow. The authors' analysis of an agent-based model shows that while feedbacks can generate the observed bimodal distribution of following, under some conditions, individual differences in tendency to be joined could also generate this pattern. This is a very interesting insight, though the specific condition when it occurs might actually not be very interesting (see my final comment below). My main problem with the paper, however, is that it leaves too much poorly explained. Thus a reader who is interested in following the model, as opposed to simply accepting the main result could have great difficulty. To me, the paper should only be accepted for publication after a major revision to make the model and the presentation of the results more comprehensible.

As stated above, we have completely restructured the presentation of the manuscript along the lines suggested by the referee. We now include much more detailed information in the presentation of the model and results, and have modified our use of figures. We hope that this now makes the presentation much clearer.

First, a small note: personality variation usually refers to consistent individual differences in behavior - consistent over time or across contexts. This model only requires that for each event, individuals differ. It is not clear that it requires that those differences persist across time or contexts.

The referee 1 is indeed correct in underlining the importance of consistency in defining personality. However, as we seek to explore the influence of personality differences on movement decisions, for each movement it is only important that there is a distribution of differences at that time (at least for the type of analysis performed here).

Figure 1b shows the distribution of proportion of 'events' involving different numbers of individuals, where the text and figure legend provide just enough information for readers to know that this represents outcomes for capuchin monkeys. I suggest describing the scenario in a bit more detail in the text; e.g., how many movements were observed? What the definition of an initiation and a join? What was the sex/age composition of the group (i.e. how much variation in leader-follower tendencies were likely given the natural history of the system)?

We added more information regarding animal movements and description of initiator/responder (see L 125-131; L150-155). We also added more information on natural history and group composition in the example of Capuchin monkeys (see L316-324; 719-729).

Equations (1) and (2) need more explanation. Lines 135-141 sort of explain these two equations, but do not do a good job. Equation 1 seems to say that the probability of a given individual j joining an

initiator depends on two parameters: α which affects the general tendency to join, and β which affects the strength of response to the number of other followers. Equation 1 has no personality variation. In contrast, equation (2) potentially has individual differences in α_i , but no feedback. Because equation (2) says that P_j depends on α_i and not α_j , it appears that there is individual variation in how strongly individuals attract others to join, but no individual differences in tendency to join? Or, because the equations are not well explained, I was not confident that this is what the authors have in mind. Later, when the authors describe Despotic and Democratic Scenarios, they appear to be saying that α_i could either represent variation in tendency to be joined (Despotic) or in tendency to join (Democratic). If this is what the authors have in mind, they need to explain this more clearly and explicitly when they first introduce equations (1) and (2). The text also mentions an α' which does not appear in either equation. The text implies that the authors may have a 3rd equation that has both the feedback and individual differences, but they do not show that equation, or is equation (2) meant to be somehow included in equation (1) in the simulation? Perhaps that is what the authors mean to say, but this is not clear.

As referee 1 says, variation was integrated in tendency to be joined (Despotic) and tendency to join (Democratic), but not both. In fact, α' from equation 3,a (PJ) can be used by taking into account the α' of the initiator (Despotic; see equation 3,b) or of the responder (equation 3,c; Democratic). We have heavily modified this section of the manuscript and now explain these equations in more detail. We hope this makes the presentation much clearer (see paragraph L120-146; L171-215).

The authors then assumed that the distribution of tendencies to attract others to join (or do they mean tendency to join) was either uniform or bimodal. I suppose this is OK, but certainly can be criticized since my sense is that more often personalities are normally distributed. Can the authors cite any references with data that distributions of tendencies to join or to attract others to join is uniform or bimodal?

Referee 1 raises a good point which illustrates the problem that there is no robust information on how personalities are distributed in a population or a group; and thus any assumption of one or the other would be arbitrary. Nonetheless, upon reflection, we agree that a Normal distribution may be more biologically realistic in the absence of any information to the contrary. We have therefore substituted our 'uniform' condition with a 'unimodal' condition, which uses a truncated normal distribution (to avoid negative values and values out of maximum range). These new analyses provide qualitatively similar response patterns to those from the uniform distribution (see Movie S1 and S2), so our conclusions are not affected by this change.

They then have equations where the initiator's probability of canceling is lower if it has more followers (a feedback mechanism), or where P_c is proportional to α_i . Some small points to clarify: In equation (3), γ is undefined – what is its biological meaning, and in equation (6), the authors should note whether α_i is Despotic or Democratic.

We now define γ in L142: 'The constant e can be understood as an amplification effect depending on the influence of the individual which has joined the movement and γ is the threshold'. Equations 4, 5 and 6 are now integrated in the text (see L250-280). Equation 6, as all the other types of PC, were applied to both, Despotic and Democratic scenarios.

Figure 2 seems to show that in their simulations, personality variation is uniform or bimodal, implying no analyses where there is NO personality variation, but I thought the authors would be comparing outcomes with vs without personality variation, where the latter would instead just have feedbacks. I also thought they would have simulations with personality variation but no feedbacks on joining, but this does not appear to be represented in figure 2.

For each scenario tested, the first set of simulations was done without personality. Thus, the first vertical line (at x axis = 0) in the previous figure 2 (for each plot) corresponded to the non-personality simulation. We have now clarified this through further explanation in the text (see L284-294; see also Fig. 7). Moreover, we added the cases with and without personality/feedback of the analytical model (previously in the supplementary) into the main text.

Figure 3 shows the Manhattan distance between simulation and experimental histogram data. While this is no doubt a useful way of quantifying fit of a model to data, the figure itself is incomprehensible. The x-axis is either Distance or Maximum, and the y-axis runs from 1-9 with axis labels that are largely unclear as to their meaning. I could guess as to what they might mean, but it would just be a guess. I can see from the color patterns that the model tended to fit the data better in some boxes than others, but how this relates exactly to the 3 brief paragraphs of text under Democratic scenario is unclear. I could probably figure it out, but it would take significant work. I am not against working to understand a paper, but it would certainly be better if the authors made this less difficult for readers. I suggest labeling every panel of figure 3 with a letter and then referring readers to compare particular panels to see the points that the authors wish to make. As a small note, the heading ‘Democratic scenario’ seems inappropriate since the results also include the Despotic scenario.

We modified previous figure 3 in order to show our results in a clearer manner. Please see the new Fig. 5 and 6.

Finally, one could view it as unsurprising, almost trivial, that if individuals differ substantially in their tendency to be joined – in particular if this tendency is bimodally distributed, then the result could be a bimodal distribution of outcomes where some individuals that tend to not be followed are not followed, while those that have a strong tendency to be followed are indeed followed. The authors note that this result is unsurprising on line 362. Fortunately, in the lines following line 362, the authors suggest some interesting ideas that go beyond their simulation’s somewhat obvious result.

While it may be somewhat predictable that bimodal character distributions produce bimodal response patterns, we also show various cases where the tendency to be followed is distributed bimodally but the final distribution of responders is not, depending on the number of leaders and type of feedback. This would suggest that the relationships are not simply trivial. We hope our new manuscript structuring makes these patterns clearer. Please see Fig. 4 and analytical model section.

Although it could be viewed as beyond the scope of this paper, it would certainly be interesting to see not only whether the different situations generate bimodal joining results as with one set of observations on capuchins, but also to report on what WERE the outcomes of other simulated scenarios? How do feedbacks and individual differences affect other outcomes of interest?

We hope our restructured manuscript goes some way toward answering these questions. For example, we have refocused the core of the manuscript to concentrate on testing for patterns of bimodality, and the comparison to the Capuchin monkeys data is supplementary to this. For all the patterns obtained during specific simulations please refer to Fig. S1, S3, S4 and S6, and Movie S1 and S2. See also the expanded analytical model section, which explores different group sizes. Unfortunately, we cannot point to a specific line or paragraph, as the results section have been heavily modified.

Reviewer #2 (Remarks to the Author):

Planas-Sitjà, Deneubourg & Cronin present a factorial modelling analysis investigating the role of personality in generating patterns of collective behaviour, and make comparisons between the patterns created through model simulations with those seen in a data set of capuchin monkey movement initiations.

At the heart of this paper is an important idea: while observed patterns of collective decision making (in particular, ‘U-shaped distributions’) have been attributed to social feedback rules, might these patterns actually arise more from variation between individuals? Investigating this hypothesis is valuable as (in my opinion at least), researchers are often too quick to attribute collective patterns to a putative social feedback rule because the two are consistent, without considering alternative explanations.

However, despite this interesting core idea, I found the paper difficult to parse and ultimately unconvincing.

The predictions of all models are compared quantitatively to one data set from a capuchin monkey study system, and yet little to no pertinent information about the study system is given when considering the plausibility of these models. For example, despotic decision systems are introduced by way of ref 36, which contains no mention of capuchin monkeys. Is there any evidence that capuchins might have a despotic system? I appreciate this might be unknown (in which case testing different models would be useful for understanding that study system), but comparison of the large factorial design against data from one study system seems ill-suited to answer wide-ranging questions about the role of personality in generating collective behaviour. As the authors note, U-shaped distributions as in Fig 1 are common across many study systems, and this is a qualitative rather than quantitative feature that models of collective decision making should reproduce. Therefore I would have found it more useful simply to know which models (and parameters) could produce a U-shaped distribution, rather than the quantitative fit to a single study system (I could not judge precisely whether the data were newly collected from a previously established system or whether the data were the same as used in a previous study). As far as I can tell, the data is acting as a proxy for the generally desirable U-shape, since the goal is to make general inferences about collective decision making, rather than to explain the specific mechanisms behind capuchin groups.

As the referee points out, there is no available information to indicate whether Capuchin monkey societies are ‘despotic’ or ‘democratic’ in organisation, and this problem is not limited to monkeys’ societies (see L91-97). It is largely for this reason that we test these two scenarios. Nonetheless, we have significantly modified the focus of our presentation to try to generalise our approach to testing simulation results against general (bimodal) expectations and not to focus only on Capuchin monkeys. We have also added various points regarding the available knowledge on the Capuchin system (see L120-130, L316-319, and methods section). We hope this new presentation thus allays the concerns raised above.

It took me a long time to work out exactly what features of the data were being compared to simulations. These are described opaquely in the manuscript, e.g. SI line 24: To compare patterns generated by simulation and empirical data’: what ‘patterns’? It would be much clearer to say exactly ‘To compare the distributions of the number of agents involved in each event’ or similar.

This comparison has been completely revised and we have endeavoured to make the presentation of these data clearer. See L75-82; Fig. 1B; L305-324.

Adding a figure showing some exemplar histograms of this distribution generated from different models would then make Fig 3 much easier to understand.

We have added a method of quantifying the bimodality of response patterns and compare this to our simulation results. We provide examples of different response patterns with reference to this index in Fig. 1b.

Although these histograms are an important feature of the data and of collective behaviour, I don't find a match between a model and the data on this one feature to be a convincing case that collective behaviours attributed to social feedback could actually be due to personality variation. There are many other ways that such systems could be explored, such as more detailed time sequences of following/abandoning, how often specific individuals are followed or lead, time intervals between decisions etc.

We agree that there are other ways to analyse the existence of social feedback, and we add a remark on that in section 3.3 of results. We use a group-level response to analyse a broad range of different scenarios. While alternative methods may also shed light on this question, the expectations of, for example, individual-level patterns over time, may be less clear and the available empirical data scarcer. We believe that our group-level analysis is robust because we can test simulation results against expected patterns and field data, and that these comparisons are valuable. We have attempted to broaden and strengthen our thesis by the addition of further analytical results in line with the comments from referee 2, and now describe how personality effects could be distinguished from social effects depending on the data available (see L564-616).

For example, it is quite clear that a despotic model with bimodal leaders should produce a U-shape, since initiators are either very strongly followed or barely followed (by definition). This could be straightforwardly tested: do initiations succeed only under specific individuals?

Indeed it can seem trivial. Nevertheless, as stated in the above response to referee 1, we find various scenarios where initiators are either superleaders (strongly followed) or superfollowers (barely followed), and bimodality is not found (see Fig 4 and S2, and analytical model section), which suggests that these patterns are not entirely predictable. While some individuals can be expected to have higher success in initiating (by definition), even "superleaders" in our simulations can cancel without success (this was observed in preliminary simulations).

This is key since these despotic-bimodal systems are predominantly those that are good fits to the data. I would have liked to see either: (i) a case made that social feedback could not be distinguished from personality variation across a wide range of possible observations or (ii) a clear delineation of what observations *can* distinguish these, as a recommendation for future empirical practice.

We show in figure 4, 5 and 6 several cases where simulated groups with and without social feedback give a similar bimodality index (our new measure of bimodality in response patterns) across a wide range of the parameter space (see also figure 7 for more detailed examples). In addition, we have introduced a new section to the analytical model part of the paper which deals specifically with delineating feedback-based and personality-based response patterns. Please see L564-616.

In conclusion, I don't think this paper represents a sufficiently substantial contribution for publication in *Communications Biology* as it focuses on a quantitative fit to one study system, and on a relatively limited view of how patterns of collective behaviour can be described.

We have modified our presentation to now test simulation results against general expectations of collective response-patterns (bimodality) and use the comparison with Capuchin monkey data as a supplement to this. We believe that the combination of almost 12 million simulations and the analytical model (corresponding to an infinity of simulations) under different scenarios and conditions in the current version provide a broad assessment of how personality might influence collective behaviour.

Minor comments:

Eqn 2, the α_i should presumably have an associated prime/apostrophe

This has been corrected.

Fig 3. The meaning of the x-axes is not given in the caption here as it is in eq. Fig S1. The colour scale is somewhat unhelpfully labelled simply as ‘values’

Figure 3 has been removed. We now give a more extensive description in the new figures.

Fig 3 and elsewhere: the scale of the alpha parameter is presumably based on the 1s intervals of the simulations, but this is very unclear, especially as one must reach L410 before seeing that this is how the simulations were done. I think alpha values should be given units (seconds?) for clarity.

We hope that this is clearer in the new version of the manuscript.

The ms notes that the results obtained using the Manhattan distance are qualitatively the same as the KS or the BC. However, it is noteworthy that differences between parameters are much clearer in Fig S2 than in Fig 3

Figure 3 was replaced by figure 6. We hope it is easier to read now.

L230 as with the point above, the scale of these alpha variation values is unclear. Furthermore, what does “variability in α ” mean? Is it standard deviation, distance between modes, range?

Variability of alpha refers to the fact that we include animal personality variation in the simulation.

The value of 1,000 refers to the actual value of alpha (x-axis of the plot when applicable). See legend figure 3 and L334.

Eqn 6 and elsewhere: the notation $Pc\alpha$ is very unclear: I think the alpha here is simply to remind us that this Pc depends on alpha prime, but it looks like a multiplicative factor.

We modified this, and now refer to this condition as $C\alpha'$ (see L270-280).

L410 Simulations: Simulating at fixed 1s intervals is probably fine, but in principle using the Gillespie algorithm would be a better approach, and much more efficient.

We appreciate the advice. For comparability with previous models we prefer to keep a time-step of 1s.

Reviewer #3 (Remarks to the Author):

In this current manuscript the authors investigate through computer modelling if differences in personality would properly describe previously published experimental data of white-faced capuchins initiating (with or without success) a collective moment of the whole group.

This dataset acquired by Petit and collaborators was previously reproduced by using an statistical model which altered the probability of a conspecific joining an initiator depending on the ratio of already decided monkeys, while taking into account that the initiator itself could abort its attempt to trigger collective movement in the group.

The authors here have implemented changes to that previously developed model, which currently takes into account differences in personalities (i.e. chances of joining an initiator or aborting an attempt). The authors have succeeded in proposing an intriguing combination of personal characteristics, which they related to different group level organizations, such as "democratic" or "autocratic" relations between the agents.

My criticism of this article is focused mainly on its form, but also on the lack of some simple (and in my opinion needed) additional analysis and comparisons of their models with trivial cases in the main text.

We have heavily revised the manuscript with the goal of improving the presentation and have added new analysis to the simulation section and analytical model. We hope the more comprehensive case now makes a more compelling case.

After checking the submission guidelines of COMMSBIO, I noticed that perhaps the suggested (or enforced?) structure of the article caused the authors to come up with a structure that is very confusing for the reader. In normal circumstances, even if the fine details of the model were left for a later methods section, a more in depth description of the model should be given at the "outline of the model section". As is, that section is just confusing and the reader has to scroll through the end of the article to read all the methods section before proceeding further.

This is indeed a correct interpretation. Based on this suggestion and other comments about manuscript clarify we have completely revised the manuscript structure and now integrate various details into results sections (see beginning of results section).

Still related to this, Eq 1 and 2 are just randomly dropped in the text, with mentions on their meaning, significance and use coming later. Unless there is a specific guideline for equations to be treated as "floating figures without a label", equations should come after a text explaining then, or at least introducing what follows. Additionally, the presentation of the previous results and modelling was lacking, and only contained a shallow description of their results. For example, the model description or results makes no mention nn how the original P_j changes over time as more agents join the initiator. Such comments are in the SI and I think they should be added to the main text.

Following the suggestions of referee 3 we modified the equation description section and we also explain the behaviour of these equations in more detail. Please see L120-146.

The shallow explanation given by the authors up to this point continues to cause problems for the reader later on: the definitions of "probability of joining" $P_j = \alpha_i$ are misleading, since from equation 2 it is assumed that an agent J joins the group solely based on the characteristic of the initiator I. While this is true for the despotic scenarios, on the democratic scenarios the equation is $P_j = \alpha_i$.

Changes to the subscript of the equation should be made to account to all possible scenarios in the study, e.g. " $P_j = \alpha_k$, where $k=i$ or j depending on despotic or democratic scenarios" and so forth.

We have modified this section and hopefully clarified this issue. Please see L171 and paragraphs L185-215.

After jumping ahead to the methods sections I was left with the current understanding of the model:

1 - An agent is sorted to become an initiator (uniform distribution? never properly described in the text)
2 - This attempt on triggering a move is active either until the initiator aborts it (T_a), or the time of the simulation runs out ($T_f=900$ time steps which is equivalent to 15min).
3 - Agents test their own probabilities P_j of joining the initiator based on their specific scenario
3b - It is not clear if the attempt succeeds in case all other agents join an attempt before T_f .

This is correct; if they reach 10 individuals the movement is successful and we abort simulation (see L162). We now provide a more detailed description of the simulations earlier in results section which hopefully makes this clear (L150 – 167).

A similar (but more in depth) analysis is given in the theoretical part of the SI, tackling how the trivial cases of the model would work, but given it's brevity and SI placement the main article suffers from not having it.

In line with this suggestion we have significantly expanded the analytical modelling section and provide a more thorough explanation of this in the main text.

Figure 3 I'm sorry to say it's abysmal. Color resolution (specially to color blind readers) makes it very hard to follow. Inverting the color scheme (such as Figure S2) would solve problems for even the most extreme cases of color blindness. The caption is lacking in information, where again SI figures have the minimal information for a reader to follow them. Furthermore, with the lack of caption, the x and y axis of each subplot are basically a mystery until the reader reads the appropriate label for figures S1 and S2.

We agree that this figure was difficult to interpret. We actually spent a great deal of time trying to decide the best way to present these data, though clearly, this was not a huge success. We have now heavily revised the presentation of our figures throughout the manuscript and hope this approach is clearer (see Fig. 6).

Overall the work is sound, the research valid, but the total experience seems like a proper manuscript was reduced to its bare minimum to fit a size limitation which I am not aware it exists. My suggestion is for drastic changes to be made in the form of the article, properly explaining the model and simulations before the results, and more importantly, properly describing the limiting cases and theoretical analysis that are only found on the SI. My review was leading to complaints about a similar analysis (concerning average time before an initiator aborts it's behavior), when I noticed similar and more in depth analysis were made in the SI but just briefly introduced in the main text, where they are sorely needed.

Referee 3's guess is totally accurate. This manuscript was transferred from Nature Communications, and we had to do a lot of cutting to fit size limitation. We have now incorporated much of the supplementary material and methods into the main text and added several new analyses. We hope that these major modifications make the manuscript easier to follow.

REVIEWERS' COMMENTS:

Reviewer #2 (Remarks to the Author):

I am broadly satisfied with the revisions made to the ms in response to all reviews, and I believe the ms is substantially improved as a result. I have two remaining issues that I do not feel the authors have fully addressed.

1. Although it is quite clear from the ms that the value of parameters such as alpha are for the specific 1 second time steps in the simulations, I continue to think it would be better to give explicit units to those parameters that are not dimensionless (for example, in Figures 3 and 5), so that simulations using an alternative framework could be easily compared.

2. The authors have more concretely established that the same collective outcome can originate from multiple mechanisms, and that certain mechanisms are not compatible with observation (eg democratic with no feedback). However, unless I missed the relevant addition, I did not see a discussion about which specific data could distinguish the remaining ambiguity (or a claim that no such data is possible). This might be data that is not always available (such as labelled individuals along with time series of decisions), but it would be nice to see that presented so that readers can take away what they would need to collect to avoid being left with the ambiguity the ms presents.

Reviewer #3 (Remarks to the Author):

I'm glad that the previous somewhat tough round of reviews has drastically improved this article. From my perspective the new representations and model description went a long way to properly describing the work and removing doubts on its details. As it stands this work presents a clear, meaningful addition to the field.

Some of the more serious complaints about triviality of the results pointed out by the previous reviewers seems to have been properly addressed and explained, and very importantly, very well exemplified by figures 3 and 4. Nevertheless, if CommsBio layout allows, a citation to the methods section in the form of a link would enable to quickly check the definition of bimodality used by the authors.

The theoretical framework of the model being introduced to the main text also did its job on clarifying some aspects of the case limits of the system. Dispelling some of the my/future reader doubts if the more trivial cases had been properly investigated.

I did catch one very minor problem in the SI though. Figures S3, S4 and S6 have the old problem of the colorbar being defined in the figure with the label "values". It should read as it is in the caption "proportions of observations".